# Utilization of Maternal Healthcare Services among Adolescent Mothers in Indonesia

**DOI:** 10.3390/healthcare11050678

**Published:** 2023-02-25

**Authors:** Ratih Virta Gayatri, Yu-Yun Hsu, Elizabeth G. Damato

**Affiliations:** 1International Doctoral Program in Nursing, Department of Nursing, College of Medicine, National Cheng Kung University, Tainan 70101, Taiwan; 2National Polytechnic of Health Bandung Ministry of Health, Republic of Indonesia, Bandung 40171, Indonesia; 3Department of Nursing, College of Medicine, National Cheng Kung University, Tainan 70101, Taiwan; 4School of Nursing, Case Western Reserve University, Cleveland, OH 44106, USA

**Keywords:** adolescent mothers, maternal healthcare, pregnancy complications, Indonesia

## Abstract

Providing maternal healthcare services is one of the strategies to decrease maternal mortality. Despite the availability of healthcare services, research investigating the utilization of healthcare services for adolescent mothers in Indonesia is still limited. This study aimed to examine the utilization of maternal healthcare services and its determinants among adolescent mothers in Indonesia. Secondary data analysis was performed using the Indonesia Demographic and Health Survey 2017. Four hundred and sixteen adolescent mothers aged 15–19 years were included in the data analysis of frequency of antenatal care (ANC) visits and place of delivery (home/traditional birth vs. hospital/birth center) represented the utilization of maternal healthcare services. Approximately 7% of the participants were 16 years of age or younger, and over half lived in rural areas. The majority (93%) were having their first baby, one-fourth of the adolescent mothers had fewer than four ANC visits and 33.5% chose a traditional place for childbirth. Pregnancy fatigue was a significant determinant of both antenatal care and the place of delivery. Older age (OR 2.43; 95% CI 1.12–5.29), low income (OR 2.01; 95% CI 1.00–3.74), pregnancy complications of fever (OR 2.10; 95% CI 1.31–3.36), fetal malposition (OR 2.01; 95% CI1.19–3.38), and fatigue (OR 3.63; 95% CI 1.27–10.38) were significantly related to four or more ANC visits. Maternal education (OR 2.14; 95% CI 1.35–3.38), paternal education (OR 1.62; 95% CI 1.02–2.57), income level (OR 2.06; 95% CI 1.12–3.79), insurance coverage (OR 1.68; 95% CI 1.11–2.53), and presence of pregnancy complications such as fever (OR 2.03; 95% CI 1.33–3.10), convulsion (OR 7.74; 95% CI 1.81–32.98), swollen limbs (OR 11.37; 95% CI 1.51–85.45), and fatigue (OR 3.65; 95% CI 1.50–8.85) were significantly related to the place of delivery. Utilization of maternal healthcare services among adolescent mothers was determined by not only socioeconomic factors but also pregnancy complications. These factors should be considered to improve the accessibility, availability, and affordability of healthcare utilization among pregnant adolescents.

## 1. Introduction

Adolescent pregnancy is an important issue around the world and around 12 million of them aged 15–19 years old experience childbirth every year especially in developing countries such as Indonesia. Based on the Indonesia Demographic and Health Survey 2017, the live birth rate for adolescent pregnancy is 36 per 1000 women [1]. Adolescent mothers are a vulnerable population because they tend to incur high-risk pregnancies that contribute to the higher rates of maternal mortality seen in developing countries. The maternal mortality rate in Indonesia is 305/100,000 live births [2]. Although the maternal mortality rate during the 2010–2017 period in Indonesia reflects a decline, this figure is still far from the 2030 Sustainable Development Goals target of 70/100,000 live births [3].

Adolescent females may be reproductively disadvantaged because their physiology is not fully developed, and their reproductive organs may not be able to fully carry out their function to sustain a prolonged labor [4]. Complications of childbirth such as anemia, hypertension, preeclampsia and eclampsia, spontaneous abortion, assisted delivery, stillbirth and gestational diabetes can occur in this vulnerable age range [5]. Furthermore, early childbearing in adolescent women is also associated with low birth-weight infants, preterm delivery, and severe neonatal complications [6]. In low- and middle-income countries, complications from pregnancy and childbirth are a leading cause of death among girls aged 15–19 years old [7].

One strategy to decrease maternal mortality is utilization of antenatal care services that monitor maternal health status during pregnancy [8]. Over half a million maternal deaths during pregnancy are due to the unavailability and poor utilization of maternal health care services around the world [9]. The use of antenatal care services provides an opportunity to promote the safety and well-being of mothers and their babies [10,11]. Accessibility, availability, and affordability of antenatal care services are still the determining factors for improving pregnancy outcomes [12]. Both the use of antenatal care services and place of delivery are two important indicators of maternal health care services.

Economic and demographic factors underlie women’s considerations for accessing maternal healthcare services in both urban and rural areas [10]. Geographic distance limits the use of healthcare services, and transportation expenses to access maternal healthcare services are a burden to pregnant adolescents [13,14].

Giving birth at home with the help of traditional birth attendants is common in developing countries [15,16,17]. A qualitative study conducted in Indonesia found that pregnant women in rural areas often prefer to use traditional birth attendants and give birth at home [18]. This suggests that financial difficulties and geographical distance may be obstacles preventing pregnant adolescents from using modern healthcare services (e.g., maternity hospitals and/or birthing centers) for antenatal and intrapartum care.

Reduction of maternal mortality rate is a main component of the 2030 Sustainable Development Goals (SDGs). As mention before, antenatal care is related to maternal mortality rate. Although several studies have explored the use of antenatal care services by adolescents in Indonesia, past studies only examined antenatal care and did not explore the place of delivery [19,20,21,22]. Evidence of specific issues facing adolescents regarding antenatal care utilization, health history, pregnancy complications, and place of delivery is scarce, particularly in developing countries such as Indonesia.

Therefore, this study aimed to investigate the utilization of maternal healthcare services and its determinants among adolescent mothers in Indonesia in which the utilization of maternal healthcare services refers to the frequency of antenatal care and place of delivery. The findings of this study are anticipated to support strategic policies and initiatives that prioritize the use of maternal healthcare services among adolescent mothers.

## 2. Materials and Methods

### 2.1. Study Design

This study utilized secondary data analysis of the Indonesian Demographic Data Survey 2017 (2017 IDHS). IDHS is part of the International Demographic and Health Survey (DHS) program conducted by the Inner-City Fund (ICF). IDHS is a nationally representative cross-sectional survey using a multistage sampling design. The 2017 IDHS was implemented under the supervision of the Health Research and Development Agency (Balitbangkes), which is one of the main units of the Indonesian Ministry of Health. The Balitbangkes performed an independent ethics review of the 2017 IDHS protocol by obtaining informed consent from all participants. Data collection interviews were conducted after informed consent was obtained.

The 2017 IDHS sample included 1970 census blocks covering urban and rural areas. The sample was nationally representative and covered the entire population residing in non-institutional dwellings in Indonesia. The census blocks obtained a household sample of 49,627 respondents who were married and aged 15–49 years (98% response rate). Permission to use the data in this study was approved by the international ICF, which is a part of the DHS program (DHS 2015).

### 2.2. Participants

Eligibility criteria for the secondary data analysis were women aged 15–19 years giving birth to their first child from January 2012 until the IDHS survey was conducted in 2017. Of the 49,627 respondents, 416 adolescent mothers met the study criteria

### 2.3. Variables

Demographic variables included maternal age, residence (rural or urban), maternal education level (primary, secondary, and above), father’s education level (primary, secondary, and above), employment status (not working or working), knowledge of pregnancy danger signs (know, do not know), baby’s birth order (first, second, and more), and insurance coverage (no, yes). The economic status quintile was regrouped into three levels: low income (<USD 200), middle income (USD 200), and high income (>USD 200). Medical history of pregnancy complications (no, yes) assessed included fever, convulsions, fetal malposition, swollen limbs, and fatigue.

Two dependent variables were used as the measure of maternal healthcare services: number of antenatal care (ANC) and place of delivery. At least four ANC visits during pregnancy were expected because the health institute in Indonesia provided four free ANC visits during the IDHS 2017 survey period. For this study, the number of ANC visits was classified into two categories: ‘less than four visits’ and ‘four visits or more’. Place of delivery was categorized as either delivery at home with a traditional birth attendant or delivery at a hospital, birthing center, public health center, or other health clinic

### 2.4. Statistical Analysis

This study used SPSS version 24.0 (IBM, Inc., Armonk, NY, USA) to analyze the data. Simple logistic regressions were performed to examine the relationship of each socio-demographic characteristic and complications of pregnancy with antenatal care visits as well as place of delivery. Multiple logistic regressions were then performed to build the models containing explanatory factors predicting the utilization of maternal healthcare services.

## 3. Results

### 3.1. Characteristics of Subjects

Of the 49,627 women in the 2017 IDHS data set, 416 met the inclusion criteria and their data were analyzed for the study (place of delivery data was missing for 4 women). Mean maternal age was 18.1 years (SD = 0.98; range 15 to 19). Approximately 7% of the participants were 16 years of age or younger, over half lived in rural areas, three-fourths were unemployed, two-thirds were low income and 43% lacked insurance coverage. The majority (93%) were having their first baby, and nearly half had no awareness of the danger signs of pregnancy complications. Around one-fourth of the adolescent mothers had fewer than four ANC visits and one-third of the adolescent mothers chose a traditional place for childbirth. Detailed descriptive statistics are provided in Table 1.

### 3.2. Bivariate Analysis Associated with ANC Visits

Table 2 details variables associated with frequency of antenatal care visits. Older age (OR 2.43; 95% CI 1.12–5.29) and low income (OR 2.01; 95% CI 1.00–4.04) were significantly associated with increased use of antenatal care visits. Adolescent mothers older than 16 years were over twice as likely to report 4 or more ANC visits versus adolescents younger than 16 years. Adolescent mothers with low income were approximately twice as likely to report 4 or more ANC visits than adolescent mothers with middle-income levels.

Pregnancy complications significantly related to frequency of antenatal care visits were fever (OR 2.10; 95% CI 1.31–3.36), fetal malposition (OR 2.01; 95% CI 1.19–3.38), and fatigue (OR 3.62; 95% CI 1.26–10.38). Adolescent mothers with fever or fetal malposition were approximately twice as likely to have 4 or more ANC visits than adolescent mothers without fever or fetal malposition. Adolescent mothers reporting pregnancy fatigue were 3.5 times more likely to have 4 or more ANC visits than adolescent mothers without pregnancy fatigue. Variables not associated with frequency of antenatal care visits were place of residence, maternal education level, paternal education, insurance coverage, parity, knowledge of pregnancy danger signs, and convulsions or swollen limbs during pregnancy.

### 3.3. Bivariate Analysis Associated with Place of Delivery

Five variables were significantly associated with delivery location including place of residence (OR 0.22; 95% CI 0.13–0.36), maternal education (OR 2.14; 95% CI 1.35–3.38), paternal education (OR 1.62; 95% CI 1.02–2.57), income (OR 2.06; 95% CI 1.12–3.79) and insurance coverage (OR 1.68; 95% CI 1.11–2.54). Adolescent mothers who lived in rural areas were less likely to choose a hospital for delivery compared to those who lived in urban areas. Adolescent mothers with at least a secondary education level were twice as likely to choose a hospital for delivery than those with a primary education level. Similarly, adolescents at a high income level were more likely to choose hospital for delivery than those adolescent mothers at a middle-income level. Adolescent mothers with insurance coverage were also more likely to choose a hospital for delivery than those without insurance coverage.

Pregnancy complications significantly related to delivery in a hospital included fever (OR 2.03, 95% CI 1.33–3.10), convulsion (OR 7.74, 95% CI 1.81–32.98), swollen limbs (OR 11.37, 95% CI 1.51–85.45), and fatigue (OR 3.65, 95% CI 1.50–8.85). Maternal age, employment status, parity, knowledge of pregnancy danger signs and fetal malposition were not associated with place of delivery. Detailed statistics are presented in Table 3.

### 3.4. Hierarchical Logistic Regression Models of Maternal Healthcare Services Utilization

Two sequential logistic regressions analyses with three steps were performed to examine the predictors of antenatal care use and choice of delivery location. First, the four demographic predictors (maternal age, place of residence, maternal education, paternal education) were added, followed by the socioeconomic predictors of income level, and insurance coverage. Pregnancy complications of fever, convulsion, fetal malposition, swollen limbs, and fatigue were added in Step 3.

In Step 1, for antenatal care use, maternal age (B = 0.89, *p* < 0.05) was the only significant demographic predictor, accounting for 2% of the variance. In Step 2, after controlling for the demographic variables, the only significant socioeconomic variable related to antenatal care use was low income level (B = 0.36, *p* < 0.05). In Step 3, after controlling for the demographic and socioeconomic variables, none of the pregnant complications significantly contributed to the variance in antenatal care. Table 4 illustrates the increase in explained variance from 5% at Step 2 to 8% at Step 3. The overall model explained only a small amount of the variance of antenatal care services (R^2^ = 0.08).

Place of residence (B = −1.41, *p* < 0.001) was a significant predictor of place of delivery in Step 1, accounting for 15% of the variance (Nagelkerke R^2^). In step 2, high income level (B = 0.80, *p* < 0.05) was a significant predictor, contributing an additional 3% of the variance. In Step 3, after controlling for the demographic and socioeconomic variables, only the pregnancy complication of fatigue (B = 1.33, *p* ≤ 0.01) was significant, accounting for 7% of the variance. Table 5 illustrates the overall model which explained 25% of the variance for place of delivery.

## 4. Discussion

The current study applied secondary data analysis to examine relationships among demographic characteristics, pregnancy complications, and utilization of maternal healthcare services among Indonesian adolescent mothers. Approximately 25% of the adolescent mothers had less than four ANC visits and one-third of the adolescent mothers chose to deliver their infants outside hospital or clinic settings. In addition, the present study reveals that adolescent mothers’ age, residence, maternal education, paternal education, income level, insurance coverage, and pregnancy complications such as fever, fetal malposition, and fatigue are related to the utilization of maternal healthcare services.

The findings of the current study show that adolescent mothers of a young age (less than 16 years old) are less likely to have ‘four or more’ ANC visits than adolescent mothers who were older than 16 years old. These findings are in concordance with one Indonesian study that adolescent mothers were less likely to have 4 or more ANC visits than young women aged 20–24 years [19]. One possible explanation for fewer ANC visits in young adolescent mothers is that younger adolescent mothers may have less awareness of the importance of antenatal care. Another possible explanation may be that younger adolescent mothers may be more dependent on an adult or others to access antenatal care [21,23].

Findings from the current study indicate that one-third of adolescent mother chose a traditional childbirth in the home. Similar to findings from previous studies, our study indicates place of delivery is influenced by where the mother lives. Indonesian adolescent mothers who live in rural areas tend to deliver a baby in the home [24]. In rural areas, culture and traditional belief are deeply rooted from generation to generation, supporting use of traditional birth attendants in the home [25]. In Indonesia, traditional birth attendants are trusted to provide complete services ranging from massaging during pregnancy, providing pregnancy advice, assisting with childbirth, and caring for the placenta as well as performing rituals during pregnancy and 40 days after giving birth. Traditional birth attendees even give spiritual service to laboring mothers by whispering a prayer or chanting incantations [26]. It should also be acknowledged that women who live in urban areas are more likely to have access to modern health facilities such as hospitals for childbirth [27].

We found that adolescent mothers who have higher education are more likely to have four or more antenatal care visits and choose a modern childbirth facility compared to those who have a lower educational background, similar to what has been previously reported [28,29]. Educational background, is well known to influence health seeking behavior [30]. Higher paternal education level was also associated with the adolescent mother’s use of four or more ANC visits and in-hospital delivery. This finding supports other studies that found paternal education influences the awareness of health needs among pregnant women and decision making related to the place of delivery [15,31]. The patriarchal system is prevalent in Indonesia; men are often expected to take on main family responsibilities and to be the breadwinner [32]; thus, they are often the dominant decision maker. Future research is needed to specifically explore the role of the husband’s decision-making regarding the utilization of maternal healthcare among pregnant adolescents.

It is not surprising that a significant relationship exists between economic status and the utilization of maternal healthcare services. This study found that adolescent women with a low income level are more than two times as likely to have four or more ANC visits than adolescent mothers with a medium income level. Those with high income are over two times more likely to use a modern place of delivery than those women with a low income level. Low income status is a deterrent to accessing modern childbirth facilities [33,34,35], whereas high income status is positively associated with delivery in a modern childbirth facility [36]. In Indonesia the average income per capita is IDR 56,000,000/year (USD 1679–3918/year) and around 10.19% of the total population is living under the poverty line [37]. The findings of this study show that those who have insurance as much as 1.68 times as likely to prefer to deliver in a modern healthcare facility than those who do not have insurance. Similar to previous reports [38], our study found that women with health insurance coverage were more likely to give birth in a modern healthcare facility than pregnant women without insurance coverage. The design of maternal healthcare services for Indonesian adolescents must include attention to low-income adolescent mothers and those living in rural areas [34,39].

We found that pregnancy complications such as fever, convulsion, swollen limbs, and fatigue significantly affect choice of delivery for adolescent mothers. Previous evidence supports a correlation of ANC use and choice of delivery location with pregnancy complications of adolescent mothers [5,40]. Fatigue is a pregnancy complication usually associated with sleep disturbances and can have serious consequences for a pregnant woman and her infant [41]. In Indonesia, fatigue is attributed to cultural factors and family customs that bind pregnant women to all the rules that must be obeyed both during pregnancy and after giving birth. This condition can be in the form of limited space for pregnant women to carry out activities outside the home, restriction of foods considered to have an adverse effect on pregnancy, and the strong belief in things beyond reason that can interfere with pregnant women and affect their pregnancy [42].

The results of this study indicate that pregnancy complications as well as demographic and socioeconomic factors determine the utilization of maternal health care services among adolescent mothers in Indonesia. Future studies are needed to examine the impact of maternal healthcare service utilization, both number of ANC visits and choice of delivery location, on maternal and infant outcomes for adolescent mothers.

The final multivariate model only accounted for 8% of the variance for antenatal care use and 25% for place of delivery choice. These findings are similar to a previous study from sub-Saharan Africa, which showed that demographic and socioeconomic variables explained 11% of the variance in antenatal care use and 30% of the variance in delivery location [28]. It is possible that other factors may contribute to the utilization of maternal healthcare services, such as awareness of pregnancy complications, health literacy, and social support. Further research is needed to investigate whether these variables are related to the utilization of maternal care services among adolescent mothers in Indonesia. However, in our hierarchy logistics regression, the Hosmer and Lemeshow tests in the last block shows ρ-value is 0.49 for antenatal care use and 0.69 for place of delivery, respectively. The findings indicate that both models fit the data.

### 4.1. Strength

The findings of this study highlight that young adolescent mothers (less than 16 years) are less likely to utilize more than four antenatal care visits and are less likely to choose to deliver their infant in a hospital or modern childbirth center. Furthermore, paternal education plays a key determinant in the utilization of maternal care services.

### 4.2. Limitations

Some limitations should be considered. First, the Indonesia Demographic Health Survey (IDHS) is carried out every five years and all existing data are obtained from self-reported questionnaires. Recall bias cannot be excluded from the survey. Second, data from the IDHS provide limited information on certain variables, particularly sexual and reproductive information. Finally, the study acknowledges four or more ANC visits as a standard, whereas the World Health Organization (WHO) has updated a new ANC guideline since 2016, in which at least eight ANC visits during pregnancy are suggested. A limitation in generalization of the study findings to those adolescent pregnant females who have at least eight ANC visits cannot be excluded.

## 5. Conclusions

This study has identified that adolescent mothers’ age, socioeconomic status, and insurance coverage are the significant determinants of utilization of maternal healthcare services in Indonesia. Young adolescent mothers limit their use of ANC visits and hospitals for delivery. Pregnancy complications such as fever, convulsion, fetal malposition, swollen limbs, and fatigue are key determinants of the utilization of maternal healthcare services among Indonesian adolescent mothers. A tailored intervention addressing economics, health resources, insurance coverage, and education is required for pregnant adolescent mothers in Indonesia.

## Figures and Tables

**Table 1 healthcare-11-00678-t001:** Demographic characteristics of adolescent mothers (*n* = 416).

Characteristic	*n* (%)
Age (Mean = 18.1, SD = 0.98)	
Maternal age (years)	
≤16	29 (7.0)
≥17	387 (93.0)
Residence	
Rural	249 (59.9)
Urban	167 (40.1)
Maternal education	
Primary	106 (25.5)
Secondary and above	310 (74.5)
Paternal education	
Primary	155 (37.3)
Secondary and above	261 (62.7)
Income level	
Low Income (<USD 200)	275 (66.1)
Middle (USD 200)	69 (16.6)
High income (>USD 200)	72 (17.3)
Maternal employment status	
Not working	327 (78.6)
Working	89 (21.4)
Parity	
1	390 (93.8)
≥2	26 (6.2)
Insurance coverage	
No	178 (42.8)
Yes	238 (57.2)
Knowledge of pregnancy danger signs	
No	180 (43.3)
Yes	236 (56.7)
ANC visits	
Less than four visits	99 (23.8)
Four visits or more	317 (76.2)
Place of delivery	
Home	142 (34.1)
Hospital or birthing center	274 (65.9)
Pregnancy complications	
Fever	
Yes	200 (48.1)
No	216 (51.9)
Convulsions	
Yes	30 (7.2)
No	386 (92.8)
Fetal Malposition	
Yes	143 (34.4)
No	273 (65.6)
Swollen Limbs	
Yes	22 (5.3)
No	394 (94.7)
Fatigue	
Yes	46 (11.1)
No	370 (88.9)

**Table 2 healthcare-11-00678-t002:** Independent variables with relation to use of antenatal care.

Variables	OR (95% CI)	*p*-Value
Demographic variables		
Maternal age (years)		
≤16 (Ref)	1	
≥17	2.43 (1.12–5.29)	0.02
Residence		
Urban (Ref)	1	
Rural	0.72 (0.45–1.16)	0.17
Maternal education		
Primary (Ref)	1	
Secondary and above	1.13 (0.68–1.88)	0.63
Paternal education		
Primary (Ref)	1	
Secondary and above	0.72 (0.41–1.25)	0.24
Socioeconomic variables		
Income level		
Low Income (<USD 200)	2.01 (1.00–4.04)	0.04
Middle (Ref) (USD 200)	1	
High income (>USD 200)	1.91 (0.97–3.75)	0.06
Maternal employment status		
Not working (Ref)	1	
Working	0.60 (0.36–1.01)	0.06
Insurance coverage		
No (Ref)	1	
Yes	1.22 (0.77–1.91)	0.39
Health History		
Parity		
1 (Ref)	1	
≥2	0.68 (0.29–1.63)	0.39
Knowledge of pregnancy danger signs		
No (Ref)	1	
Yes	0.74 (0.34–1.61)	0.73
Pregnancy complications		
Fever		
No (Ref)	1	
Yes	2.10 (1.31–3.36)	<0.01
Convulsions		
No (Ref)	1	
Yes	2.98 (0.88–8.67)	0.07
Fetal Malposition		
No (Ref)	1	
Yes	2.01 (1.19–3.38)	<0.01
Swollen Limbs		
No (Ref)	1	
Yes	6.95 (0.92–52.36)	0.06
Fatigue		
No (Ref)	1	
Yes	3.63 (1.27–10.38)	0.01

**Table 3 healthcare-11-00678-t003:** Independent variables with relation to place of delivery.

Variables	OR (95% CI)	*p*-Value
Demographic variables		
Maternal age (years)		
≤16 (Ref)	1	
≥17	0.88 (0.39–2.00)	0.77
Residence		
Urban (Ref)	1	
Rural	0.22 (0.13–0.36)	<0.001
Maternal education		
Primary (Ref)	1	
Secondary and above	2.14 (1.36–3.38)	0.001
Paternal education		
Primary (Ref)	1	
Secondary and above	1.62 (1.02–2.57)	0.04
Socioeconomic variables		
Income level		
Low Income ( < USD 200)	1.47 (0.82–2.61)	0.18
Middle (Ref) (USD 200)	1	
High income (>USD 200)	2.06 (1.12–3.79)	0.02
Employment status		
Not working (Ref)	1	
Working	0.75 (0.46–1.22)	0.25
Insurance coverage		
No (Ref)	1	
Yes	1.68 (1.11–2.53)	0.01
Health History		
Parity		
1 (Ref)	1	
≥2	0.79 (0.35–1.79)	0.58
Knowledge of danger signs		
No (Ref)	1	
Yes	0.82 (0.44–1.54)	0.54
Pregnancy Complications		
Fever		
No (Ref)	1	
Yes	2.03 (1.33–3.10)	0.001
Convulsions		
No (Ref)	1	
Yes	7.74 (1.81–32.98)	<0.01
Fetal Malposition		
No (Ref)	1	
Yes	1.50 (0.96–2.34)	0.07
Swollen Limbs		
No (Ref)	1	
Yes	11.37 (1.51–85.45)	0.02
Fatigue		
No (Ref)	1	
Yes	3.65 (1.50–8.85)	<0.01

**Table 4 healthcare-11-00678-t004:** Hierarchical logistic regression models for antenatal care use.

	Model 1			Model 2			Model 3		
	B	OR	CI 95%	B	OR	CI 95%	B	OR	CI 95%
Maternal Age (years) (Ref: ≤ 16)	0.89	2.43 *	(1.12–5.29)	0.99	2.70 **	(1.22–5.97)	0.96	2.61 *	(1.16–5.87)
Low Income (Ref: Middle income)				0.36	2.11 *	(1.04–4.26)	0.67	1.96	(0.96–4.01)
Fever (Ref: No)							0.28	1.32	(0.61–2.86)
Fetal Malposition (Ref: No)							0.27	1.32	(0.58–3.0)
Fatigue (Ref: No)							1.01	2.75	(0.89–8.47)
Hosmer and Lemeshow test			0.10			0.21			5.43
Sig			0.01			0.89			0.49
Nagelkerke R^2^			0.02			0.05			0.08

Significance and 95% confidence levels based on odds ratios; * *ρ* < 0.05; ** *ρ* < 0.01.

**Table 5 healthcare-11-00678-t005:** Hierarchical logistic regression models for place of delivery.

	Model 1			Model 2			Model 3		
	B	OR	CI 95%	B	OR	CI 95%	B	OR	CI 95%
Residence (Ref: Urban)	−1.41	0.24 ***	(0.14–0.40)	−1.48	0.22 ***	(0.13–0.38)	−1.54	0.23 ***	(0.12–0.37)
Maternal education (Ref: Primary)	0.48	1.62	(0.93–2.81)	0.37	1.45	(0.83–2.55)	0.27	1.31	(0.73–2.33)
Paternal education (Ref:Primary)	0.24	1.28	(0.75–2.18)	0.18	1.20	(0.69–2.08)	0.15	1.16	(0.66–2.05)
High income (Ref: Middle income)				0.80	2.24 *	(1.13–4.43)	0.78	2.19 *	(1.07–4.46)
Insurance coverage (Ref: No)				0.44	1.56	(0.97–2.54)	0.34	1.40	(0.86–2.29)
Fever (Ref: No)							−0.18	0.82	(0.48–1.41)
Convulsions (Ref: No)							1.43	4.18	(0.89–19.5)
Swollen Limbs (Ref: No)							1.95	7.06	(0.83–59)
Fatigue (Ref: No)							1.33	3.78 **	(1.29–11)
Hosmer and Lemeshow test			1.76			16.26			5.55
Sig			0.88			0.03			0.69
Nagelkerke R^2^			0.15			0.18			0.25

Significance and 95% confidence levels based on odds ratios; * *ρ* < 0.05; ** *ρ* < 0.01; *** *ρ* < 0.001.

## Data Availability

This study used data sets available from USAID’s Demographic and Health Survey (DHS) program. After registration on the website, data sets can be downloaded and used via the DHS program website: https://dhsprogram.com/data/new-user-registration.cfm. Accessed on 10 January 2020.

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
