# Peer review of "Utilization of Maternal Healthcare Services among Adolescent Mothers in Indonesia"

_healthcare, 2023, doi:10.3390/healthcare11050678_

Round 1

Reviewer 1 Report

This manuscript address the topic of the utilization of maternal healthcare services among adolescent mothers in Indonesia. This paper describes the characteristics of mothers in this situation relative to their decision of using or not maternal healthcare services. Besides, they propose a predictive model to better understand the profile of mothers using that services. Results reported in this paper might be informative for policy makers interested in promoting the use of healthcare among this population. However, some aspects should be addressed before this paper is ready for publication:

Authors should explain in their introduction why could be relevant to study this topic. Further, no clear hypotheses or aims are formulated that help the reader to understand the study design and analyses approach.

Regarding discussion section, some explanations provided by authors lack a reference that support what they propose (e.g. lines 259-263).

Authors should also address the limitations of their study in the discussion section.

Author Response

Dear Editor-in-Chief and Reviewers 

Thank you for the valuable comments and suggestions from reviewers. We have revised the manuscript accordingly and given red font color to mark the revised parts please see the attachment.

We look forward to the possible publication in MDPI Healthcare.

Sincerely yours

Yu-Yun Alice Hsu, RN, Ph.D

Reviewer 2 Report

Dear Authors

With all respect, the manuscript was so interesting and impressive in term of informative content for healthcare management in Indonesia and would be applied for that, but in my opinion is not acceptable and related to this Journal. I recommend to submit to another journals after minor revision in presentation of results.

Regards

Author Response

Dear Editor-in-Chief and reviewers

Thank you for the valuable comments and suggestions from reviewers. We have revised the manuscript accordingly and given red font color to mark the revision parts please see the attachment.

We look forward to the possible publication in MDPI Healthcare.

Sincerely Yours

Yu-Yun Alice Hsu, RN, Ph.D

Reviewer 3 Report

Thank you for giving me the chance to review this work, which I found interesting to read. The manuscript is well drafted, however, I have a minor comments for improvement: 

Line 85: do you mean it was a retrospective study that utilised secondary data analysis etc?

Abstract and line 132: "7% of the participants were 16 years of age or younger" I am not sure why the emphasis is on the minority of the participants? can you express it differently? e.g., the majority were in ?? age group.

Line 138: Do you mean detailed demographic information of the participants are provided in Table 1?

In table 1: n & % could be merged in one column as n(%)

In table 1: how did you determine the middle income level? it is not clear. 

Line 298: :.... in Indonesia. In Indonesia, 10.9% ........" The two sentences could be merged in one sentence, with removing the repetitive word "Indonesia".

Author Response

Dear Editor-in-Chief and reviewers

Thank you for the valuable comments and suggestions from reviewers. We have revised the manuscript accordingly and given red color to mark the revision parts. Please see the attachment. We look forward to the possible publication in MDP Healthcare.

Sincerely Yours,

Yu-Yun Alice Hsu, RN, Ph.D

Round 2

Reviewer 2 Report

Dear Authors

Thanks for your valuable manuscript.

Title was good and clear.

Abstract was well structured. Introduction was complete and well written.

Method was well designed with pretty good statistical analysis.

Result was well reported and sound and clear.

Discussion was well written and interpreted inconsistent with result.

Conclusion was inconsistent with discussion and result.

Regard